# The Shared Experience Actor–Critic (SEAC) Approach for Allocating Radio Resources and Mitigating Resource Collisions in 5G-NR-V2X Mode 2 Under Aperiodic Traffic Conditions

**DOI:** 10.3390/s24206769

**Published:** 2024-10-21

**Authors:** Sawera Aslam, Daud Khan, KyungHi Chang

**Affiliations:** Department of Electrical and Computer Engineering, Inha University, Incheon 22212, Republic of Korea; sawerajutt@inha.edu (S.A.); daudkhan@inha.edu (D.K.)

**Keywords:** 5G-NR-V2X Mode 2, SB-SPS, DS, reinforcement learning, SEAC, aperiodic traffic

## Abstract

5G New Radio (NR)-V2X, standardized by 3GPP Release 16, includes a distributed resource allocation Mode, known as Mode 2, that allows vehicles to autonomously select transmission resources using either sensing-based semi-persistent scheduling (SB-SPS) or dynamic scheduling (DS). In unmanaged 5G-NR-V2X scenarios, SB-SPS loses effectiveness with aperiodic and variable data. DS, while better for aperiodic traffic, faces challenges due to random selection, particularly in high traffic density scenarios, leading to increased collisions. To address these limitations, this study models the Cellular V2X network as a decentralized multi-agent networked Markov decision process (MDP), where each vehicle agent uses the Shared Experience Actor–Critic (SEAC) technique to optimize performance. The superiority of SEAC over SB-SPS and DS is demonstrated through simulations, showing that the SEAC with an N-step approach achieves an average improvement of approximately 18–20% in enhancing reliability, reducing collisions, and improving resource utilization under high vehicular density scenarios with aperiodic traffic patterns.

## 1. Introduction

The evolution of intelligent transportation systems (ITSs) has increasingly underscored the importance of integrating connected and automated driving technologies, with V2X communications emerging as a critical enabler. V2X technology has garnered significant attention from both academic and industrial sectors, as it plays a pivotal role in enhancing road safety and transportation efficiency [1]. Within this domain, Cellular Vehicle-to-Everything (C-V2X) technology, encompassing long-term evolution (LTE) and fifth-generation New Radio (5G-NR) standards, facilitates the seamless information exchange between vehicles, roadside units (RSUs), and vulnerable road users (VRUs) [2,3,4].

The initial C-V2X specifications, introduced in 3GPP Release 14 (Rel-14), featured two operational Modes: the network-based Mode 3, which utilizes the LTE-Uu interface for vehicle-to-network communication, and direct Mode 4, which enables device-to-device communication via the PC5 interface. Despite its advantages, LTE-V2X does not fully satisfy the stringent requirements for autonomous driving, particularly those associated with ultra-reliable low-latency communication (URLLC). In response, 3GPP initiated the standardization of 5G NR in two phases; Phase 1 (Rel-15) focused on enhanced mobile broadband (eMBB) and preliminary URLLC studies, while Phase 2 (Rel-16 onwards) emphasized URLLC and network performance enhancements. 5G-NR-V2X, which extends beyond broadcast transmissions, supports both unicast and multicast transmissions. Like C-V2X, 5G-NR-V2X operates in two transmission Modes: Mode 1 (centralized) and Mode 2 (decentralized). In Mode 1, resource scheduling is managed by the base station (enB) when vehicles are within coverage areas, whereas in Mode 2, vehicles autonomously reserve resources using the SB-SPS algorithm in out-of-coverage regions. 5G-NR-V2X Mode 2 introduces a mandatory re-evaluation mechanism that includes both the SB-SPS and DS schemes. The SB-SPS scheme is inherited from LTE-V2X with minor modifications, while the DS scheme is a new reservation-less approach that requires selecting new transmission resources for each generated message [1,5,6].

To date, only a few studies have examined the performance of the SB-SPS scheme in 5G-NR-V2X Mode 2, revealing that increased vehicular traffic density and aperiodic traffic with varying data sizes lead to inefficiencies in the SB-SPS method. These inefficiencies result in poor radio resource utilization (RRU) and a higher probability of resource collisions, particularly when a vehicle fails to transmit a message in an intended slot due to a collision. To address these limitations, novel artificial intelligence (AI)-based solutions and alternative techniques have been proposed. Conversely, there has been limited analysis of the DS scheme. Machine learning has proven invaluable across various domains, including vehicular communications, offering optimal solutions through advanced algorithms. Several studies have employed machine learning to address challenges in V2X communications, such as optimizing resource allocation in C-V2X Mode 4 to mitigate network contention and reducing the signaling overhead in C-V2X Mode 3 [5]. Cutting-edge AI techniques, including machine learning and deep learning, are providing innovative solutions to wireless communication challenges, particularly through the application of deep neural networks in supervised, unsupervised, and reinforcement learning settings [6,7].

Despite the extensive efforts to apply machine learning techniques to address the challenges in 5G-NR-V2X Mode 2, achieving significant breakthroughs remains challenging [8,9,10]. This paper presents a novel approach to resolving persistent collision issues in 5G-NR-V2X Mode 2 by implementing the SEAC reinforcement learning (RL) technique. Our research exclusively focuses on optimizing Mode 2 communication within the European V2X framework, particularly for Cooperative Awareness Messages (CAMs) and Collective Perception Messages (CPMs), which are safety-critical and transmitted more frequently than other message types. Our objective is to reduce collisions during resource reselection and to enhance packet reception ratios.

The remainder of this paper is organized as follows: Section 2 reviews the literature on SB-SPS and DS, as well as related work. Section 3 introduces the proposed SEAC technique, and the methodology employed. Section 4 describes the simulation environment, Section 5 presents the results and their analysis, and Section 6 concludes the paper.

The main findings of this paper are as follows:The performance of the SB-SPS scheme can significantly deteriorate in the presence of aperiodic traffic, especially with increasingly stringent packet delay budget (PDB) constraints. However, it remains the optimal solution for serving fixed-size periodic traffic, provided that the reservation periodicity aligns with the traffic generation period;Conversely, the performance of the DS scheme remains independent of PDB requirements, making it most effective for handling both fixed- and variable-size aperiodic traffic under low or average density conditions;Although the DS scheme performs better than SB-SPS in aperiodic traffic scenarios, its effectiveness diminishes as traffic density increases due to the random selection of transmission resources, leading to higher collision probabilities;The SEAC strategy consistently outperforms both SB-SPS and DS schemes, particularly in aperiodic traffic scenarios, by improving reliability, reducing collisions, and enhancing resource utilization.

## 2. Overview of 5G-NR-V2X Mode 2

In 5G-NR-V2X Mode 2, communication systems utilize the physical (PHY) layer for signal transmission and the medium access control (MAC) layer for managing access to the communication medium. The 3GPP also devoted significant efforts to the MAC sublayer to develop a new resource allocation Mode, known as 5G-NR-V2X Mode 2. The MAC layer employs two main resource allocation schemes: SB-SPS and DS.

SB-SPS operates by allowing vehicles to periodically reserve subchannels for a fixed number of transmissions, defined by the reselection counter $C_{resel\;}\;$ and the resource reservation interval (RRI). Vehicles using the SB-SPS scheme periodically reserve the selected subchannel(s) for a number of $C_{resel}\;$ consecutive transmissions. The time between consecutive reservations is called the resource reservation interval (RRI), and 5G-NR-V2X Mode 2 supports any RRI value in the {0, [1:1:99], [100:100:1000]} ms range, where in the [x:y:z] notation, x denotes the minimum allowed value; z, the maximum; and y, the incremental step between consecutive values. Vehicles employ the SCI to broadcast the adopted RRI configuration and to inform neighboring vehicles about their next reservation. Depending on the selected RRI, the value of the $C_{resel}$ is set as follows: if RRI ≥ 100 ms, $C_{resel}$ is randomly set between the [5, 15] interval [1,5].

After every transmission, the value of the $C_{resel}$ is decremented by one, and when $C_{resel}$ = 0, the ego vehicle selects new subchannels with probability 1 − P, where P is the keep probability; P ∈ [0, 0.8]. On the other hand, vehicles using the DS scheme select new subchannels every time a new message is generated, and they are not allowed to place any reservations. In other words, DS is the reservation-less variant of the SB-SPS scheme, which sets $C_{resel}$ = 1 and P = 0. Despite being characterized by a totally different reservation policy, the SB-SPS and DS schemes employ the same resource reselection process (illustrated in Figure 1). The ego vehicle’s subchannel selection process involves two main phases: the sensing window and the selection window, as shown in Figure 1. During the sensing window, which spans from $n-T_{o}$ to $n-T_{proc}$ depending on the subcarrier spacing (SCS), the ego vehicle identifies and excludes subchannels reserved by neighboring vehicles if the reservation signal exceeds a certain Reference Signal Received Power (RSRP) threshold. Additionally, subchannels previously used by the ego vehicle during this period are excluded due to half-duplex limitations.

In the selection window, which extends from $n-T_{1}$ to $n-T_{2\;}$ based on the SCS and the PDB, the ego vehicle compiles a list of candidate subchannels. If the list does not cover at least a set percentage (β%) of the window, the RSRP threshold is increased by 3 dB, and the process is repeated. Depending on the priority of the TB, β can be set to 20, 35, or 50. Once an adequate list is determined, the ego vehicle randomly selects adjacent subchannels to accommodate its transmission needs.

However, SB-SPS has notable limitations:Persistent collisions: Periodic reservations can result in repeated collisions if multiple vehicles reserve the same resources;Inefficiency with variable traffic: SB-SPS struggles to accommodate aperiodic or variable-sized data, leading to unutilized reservations and higher collision probabilities;Latency issues: The fixed periodic nature of SB-SPS does not adapt well to varying latency requirements, causing delays for latency-sensitive applications. Figure 2 illustrates these problems in the SB-SPS resource selection process, showing how issues like size reselection, latency reselection, and packet interval reselection lead to inefficiencies.

DS in vehicular networks allows vehicles to randomly select subchannels for each transmitted message without prior reservations, assuming that all subchannels within the selection window are available, as shown in Table 1. This method, like a multichannel slotted Aloha strategy, operates independently of the selection window’s width and PDB. The collision probability under DS is primarily influenced by the message generation rate and size, making it an efficient approach that minimizes collision risks, particularly when only a subset of vehicles adopts this scheme.

DS on the other hand, involves the random selection of new subchannels for each transmission without pre-reserving resources. This scheme better handles aperiodic and variable traffic but faces the following significant challenges:High collision probability: Random selection increases collision risk, especially in dense traffic scenarios;Lack of coordination: Real-time resource selection can lead to inefficient subchannel usage and higher interference;PDB issues: DS does not explicitly consider the PDB for individual packets, potentially missing strict latency requirements for delay-sensitive applications, leading to delays and suboptimal performance.

The current landscape of research on resource allocation strategies in 5G-NR-V2X Mode 2, as summarized in the comparison in Table 2, highlights significant efforts to address the limitations of the SB-SPS and DS techniques, particularly in handling aperiodic traffic and periodic traffic. In Table 2, a cross (✗) indicates that the technique is not included in the respective study, while a checkmark (✓) signifies that it is included. Various approaches, including Multi-Agent Actor–Critic (IAC, SEAC) [2], deep reinforcement learning (Q-learning) [5,9,10], and SB-SPS with lookahead mechanisms [11,12,13], have been explored to improve reliability, reduce collisions, and enhance resource utilization in V2X communications. However, despite these advances, no existing work has fully addressed the challenges associated with SB-SPS in aperiodic traffic, nor has there been substantial exploration of DS, especially under conditions of high traffic density where the likelihood of collisions increases due to the random resource selection inherent in DS [14,15].

This work addresses these gaps by proposing a SEAC reinforcement learning technique that not only overcomes the limitations of SB-SPS but also enhances the reliability and resource utilization in aperiodic environments. As demonstrated by the results, SEAC significantly outperforms both SB-SPS and DS, particularly in scenarios where the traffic density is high, proving to be a superior method for reducing collisions and optimizing resource allocation.

## 3. System Design and the Proposed SEAC-Based Resource Allocation Methodology

In this work, we focus specifically on aperiodic traffic in 5G-NR-V2X Mode 2, which presents unique challenges in resource allocation. Traditional methods like SB-SPS and DS have limitations in this context. SB-SPS struggles with variations in message size, packet generation latency, and the intervals between packet generations, leading to underutilized resources and frequent collisions. On the other hand, while DS is more flexible in handling aperiodic and variable-sized traffic, it has a high collision probability in environments with dense vehicular traffic. Moreover, DS does not consider packet delay budget (PDB) constraints, which are crucial for latency-sensitive applications such as autonomous driving and real-time video streaming [16,17].

To address these challenges, we propose the Shared Experience Actor–Critic (SEAC) reinforcement learning technique, which optimizes resource allocation in aperiodic traffic scenarios by overcoming the limitations of both SB-SPS and DS. SEAC leverages shared experiences among vehicular agents to reduce collisions and to improve resource utilization. It dynamically allocates resources by considering both real-time decisions and long-term impacts, ensuring sustained performance in high-density vehicular environments [18].

Figure 3 provides a general framework that illustrates both the existing (DS and SB-SPS) and the proposed SEAC techniques for a better understanding. While this figure includes a comparison of the traditional and proposed methods, the main focus of our work is on aperiodic traffic and how SEAC enhances resource allocation in such scenarios. The decision-making process shown in the figure is based on the traffic type (aperiodic vs. periodic), with SEAC addressing the challenges specifically associated with aperiodic traffic.

In the 5G-NR-V2X Mode 2 network, the radio resource allocation task is addressed through a fully decentralized multi-agent networked MDP. Vehicular agents operate in a shared road traffic environment, constantly exchanging CAMs and CPMs. These messages provide critical information such as position, distance to neighboring vehicles, and forecasts of object locations. This decentralized model bases its actions on each agent’s local observations and assigns rewards specific to each agent’s context. The primary objective for all agents is to ensure the successful transmission of V2X messages within the selected radio resources while avoiding collisions with other agents. By employing experience-sharing techniques, the agents enhance their learning capabilities, gaining valuable insights from the actions and outcomes experienced by their peers [18]. Our approach utilizes gated recurrent unit (GRU) layers to process and forecast the state information, optimizing resource allocation decisions and improving the overall network efficiency. To formulate an MDP, the components—state, action, reward, and experience—must be clearly defined in the context of radio resource allocation.

### 3.1. Network State Representation

In 5G-NR-V2X Mode 2 networks, the state includes essential information for each vehicle’s decision making regarding radio resource allocation. Accurate observation and processing of these parameters are crucial for efficient collision-free communication in vehicular networks.

The state $s_{t\;}$ at a given time t includes the following components:(1)$$s_{t\;}=\left\{C^{k\times m},\;\Delta d,\;I\_sf\right\}$$

#### 3.1.1. Distance to Neighboring Vehicles (Δ*d*)

This parameter represents the distance between the ego vehicle and its neighboring vehicles. This is crucial for the reuse of radio resources, as distant vehicles can use the same resources without causing interference

#### 3.1.2. Forecast Indicator of the Possibility of an Object (*I_sf*)

This indicator reflects the likelihood of aperiodic traffic within the vehicle’s sensor field of view (FOV). Derived from Cooperative Perception Messages (CPMs), it helps in predicting potential traffic events and adjusting resource allocation.

#### 3.1.3. Radio Resource Occupancy ($C^{k\times m}$) in the Previous 1 s Time Window

This parameter indicates the number of resource blocks (RBs) occupied in each subchannel *k* and timeslot *m* within the selection window $T_{sel\;}$ [19,20].

The number of resource blocks occupied in subchannel k and timeslot m are given by
(2)$$C^{k\times m}\;=\;\sum_{k\;\in \;K}\;\sum_{m\;\in \;M}\;RB_{s}$$
where *k* is the set of subchannels, and *m* is the set of timeslots.

We capture the temporal variations of these parameters over the previous time window using a GRU-based recurrent neural network (RNN). These components are captured and processed in the system using gated recurrent units (GRUs). The GRUs track temporal variations in these parameters and use them as multi-dimensional inputs to predict future states, aiding in more informed decision making regarding resource allocation.

### 3.2. Resource Management Decisions

The action space for each agent in our resource scheduler is defined by the available radio resource pool, denoted as $C^{k\times m},$ where k represents the number of subchannels (each consisting of groups of resource blocks (RBs)), and m represents the number of slots. Furthermore, the total channel bandwidth is 20 MHz, operating in the 5.9 GHz ITS band. We also incorporate a reselection counter $\;C_{Resel}$ as per ETSI specifications, with a range uniformly distributed between 5 and 15 [20]. This counter determines the duration for which a vehicle holds the selected resources for future transmissions. The system operates under design constraints such as channel bandwidth (20 MHz), V2X transmission power (0 to 23 dBm), and an RSRP threshold of −128 dBm. These constraints ensure that radio resources are efficiently allocated and that the signal strength is sufficient for reliable communication.

Each vehicle, with a mean speed of 140 km/h (and standard deviation of 3 km/h), selects one radio resource from $C^{k\times m}$ and one category of the $C_{Resel}$ value. The action selection mechanism aims to achieve the following:Select a radio resource and decide the duration for using the same resource for future transmissions, based on the V2X data traffic pattern forecasted by the GRU layers;Detect potential resource collision risks and re-evaluate the resource selection within the PDB;Avoid scheduling the same resource for subsequent transmissions after detecting collisions in one of the intended slots and inform other agents to avoid the same resource;Adapt $C_{Resel}\;$ based on the V2X data traffic pattern to ensure that reserved resources are utilized efficiently.

The actor network selects a radio resource based on the softmax probability values. The critic network evaluates the impact of the selected action by computing the state value after the environment transitions to a new state. This continuous evaluation by the critic network provides feedback to the actor network, helping it to refine its policy and make more effective resource allocation decisions.

### 3.3. Performance Feedback Mechanism

The critic component in our model plays a crucial role in evaluating the actions taken by the actor within the N-step actor–critic framework. It provides a value function estimate, which helps in assessing the quality of the actions in terms of expected rewards. The critic network receives the current state and action as inputs and outputs the value of the resulting state.

#### 3.3.1. Immediate Reward

The reward $r_{t\;}$ for each agent i  at time t is calculated based on the following criteria:
The reward $r_{i}\left(a_{t}^{i}/s_{t}^{i}\right)\;$= $S_{txs\;}^{i}$ if no collisions are detected and the resource allocation latency is $L_{RA}^{i}$ $\leq \;L_{PDB};$The reward $r_{i}\left(a_{t}^{i}/s_{t}^{i}\right)=S_{txs\;}^{i}-S_{re\;}^{i}$ if collisions are detected but successfully re-evaluated within $L_{PDB}$;The reward $r_{i}\left(a_{t}^{i}/s_{t}^{i}\right)=-F_{txf\;}^{i}$ if the agent cannot reselect resources within $L_{PDB}$ due to multiple neighboring vehicles choosing the same resource.
(3)$$r_{i}\left(a_{t}^{i}/s_{t}^{i}\right)=\left\{\;\;\begin{array}{c} \;\;\;\;\;\;\;\;\;\;\;\;\;\;\;\;\;\;\;\;\;\;\;\;\;\;\;\;\;\;\;\;L_{RA}^{i}\;\leq \;\;L_{PDB}\\ \;S_{txs\;}^{i}-S_{re\;}^{i}\;\;\;\;\;\;\;\;\;\;\;\;\;\;L_{RA}^{i}\;\leq \;\;L_{PDB}\\ \;\;\;\;\;\;\;\;\;\;\;\;\;\;\;\;\;\;\;\;\;\;\;\;\;\;\;\;\;\;\;\;L_{RA}^{i}>\;L_{PDB} \end{array}\right.$$where
$S_{txs\;}^{i}\;$ is the number of successful transmissions for the agent;$S_{re\;}^{i}\;$ is the number of re-evaluated successful transmissions;$F_{txf\;}^{i}$ is the number of failed transmissions;$L_{RA}^{i}\;$ is the resource allocation latency;$L_{PDB}\;$ is the packet delay budget.

#### 3.3.2. Cumulative Rewards in the N-Step Actor–Critic Framework

The N-step actor–critic framework computes the cumulative reward $G_{t}$ over *N* steps to balance immediate rewards and long-term performance.
(4)$$G_{t}=\sum_{k=0}^{N-1}\gamma^{k}r_{t+k}+\gamma^{N}V\left(s_{t+N}\right)$$where$G_{t}$ is the cumulative reward at time step t;γ  is the discount factor;$r_{t+k}\;$ is the immediate reward at time step t+k;$V\left(s_{t+N}\right)\;$ is the estimated value of the state at time step t+N;*N* represents the number of steps used in the *N*-step actor–critic method, where cumulative rewards are calculated over *N* time steps to balance between immediate and future rewards.

The immediate rewards provide a short-term measure of success for each action taken, focusing on the success of individual transmissions and collision avoidance. These rewards are accumulated over N steps to form the cumulative reward $G_{t}$, which evaluates the long-term impact of actions on the resource allocation efficiency and overall network performance. This dual approach ensures that the model not only makes effective immediate decisions but also optimizes for future states.

### 3.4. Cumulative Learning Process

Each vehicular agent takes independent actions based on its local observations of the environment. These actions are influenced by the agent’s policy, which is continuously refined through interactions with the environment. After executing each action, the agent receives a reward (either positive or negative) that reflects the immediate impact of its decision.

The sequence of states, actions, and rewards generated by the agent over multiple time steps constitutes an ”on-policy” trajectory, which is referred to as experience. This experience captures the agent’s interactions with the environment and is crucial for updating the policy and value functions, ensuring that the agent adapts to changing network conditions and traffic patterns

State transitions: The changes in the agent’s state after taking actions, such as moving from one radio resource to another;Actions: The specific radio resource selections and reselection decisions made by the agent;Immediate rewards: The rewards received based on the success of transmissions, collisions, and resource allocation latencies;Cumulative rewards: The cumulative impact of actions over multiple time steps, which are used to update the agent’s policy.

### 3.5. SEAC-Based Resource Allocation Algorithm

In this section, we detail the SEAC methodology for efficient resource allocation in 5G-NR-V2X Mode 2 networks. The SEAC approach leverages a multi-agent setup, as depicted in Figure 4, where each vehicular agent learns to optimize its resource allocation strategy while sharing experiences with other agents to enhance the overall system performance.

The SEAC model efficiently allocates communication resources among vehicles by employing both actor and critic networks. The actor network is responsible for selecting actions based on the current state $s_{t}$ provided by the 5G-NR-V2X Mode 2 environment. The actor network generates an action $a_{t}$ using its online policy network $\mu \;\left(s_{t},\theta^{\mu }\right)$, which is subsequently executed in the environment. This action influences the environment, leading to a new state $s_{t+1}$ and a corresponding reward $r_{t\;}$ [21,22].

The critic network evaluates the selected actions by computing the value function, providing feedback to the actor network. It consists of two *Q*-values: the online network and the target value network. The online value network calculates the *Q*-value $Q\;\left(s,\;a,\;\theta^{Q}\right)$, while the target value network computes the target *Q*-value $Q^{\prime }\left(s_{t+1},\;a^{\prime },\;\theta^{Q^{\prime }}\right)$, where $a^{\prime }=\mu^{\prime }\left(s_{t+1}\right)\;$ helps in refining the value function

Figure 4 illustrates the Shared Experience Actor–Critic (SEAC) mechanism, where vehicles collaborate to improve resource allocation through shared learning in a 5G-NR-V2X environment. The process begins with each vehicle observing its environment state and using the actor network to select a resource allocation action. The critic network then evaluates this action by calculating a *Q*-value that reflects the effectiveness of the resource allocation. The outcome (state–action–reward tuple) is stored in a local experience pool for future training. To further enhance learning, vehicles share their experiences via V2V communication, which are aggregated into a shared experience pool, enabling each vehicle to learn from both its own and others’ experiences. This shared learning improves the overall decision-making and resource optimization process.

The actor and critic networks are updated through a process involving gradient calculations (policy gradient for the actor and Q-value gradient for the critic) based on the critic’s feedback. Soft updates are applied to ensure stable learning, adjusting the target policy and value networks gradually. The temporal difference (TD) error is computed to refine the critic’s evaluations, ensuring accurate learning over time. This iterative process allows vehicles to continuously refine their policies, leading to more efficient resource allocation and improved network performance by leveraging both local and shared experiences.

The TD error is crucial in evaluating the difference between the expected value of the current state–action pair and the value after observing the outcome. This error helps in refining the value function.

The TD error can be expressed as
(5)$$\;\;\;\;\;\;\;\delta_{t}=r_{t}+\gamma Q^{\prime }\left(s_{t+1},\;a^{\prime },\theta^{Q^{\prime }}\right)-Q\;\left(s_{t},\;a_{t},\;\theta^{Q}\right)$$
where

$r_{t}$ is the immediate reward at time *t*;γ is the discount factor;$Q^{\prime }\left(s_{t+1},\;a^{\prime },\theta^{Q^{\prime }}\right)\;$ is the target value for the next state–action pair;$Q\;\left(s_{t},\;a_{t},\;\theta^{Q}\right)$ is the current estimate of the state–action value.

The critic’s value function $Q\;\left(s,\;a,\;\theta^{Q}\right)$ is updated based on the TD error, and the actor’s policy is refined using the policy gradient calculated by the critic. Soft updates are applied to the target policy network and the target value network, ensuring smooth transitions and stable learning by gradually moving the parameters toward those of the online networks.

To ensure training stability, the parameter update for the critic’s value network is given by
(6)$$\theta^{Q}=\;\theta^{Q}+\alpha \delta_{t}\nabla_{\theta_{Q}}Q\left(s,\;a,\;\theta^{Q}\right)$$
where

α is the learning rate;$\nabla_{\theta_{Q}}Q\left(s,\;a,\;\theta^{Q}\right)$ represents the gradient of the value function with respect to its parameters.

Similarly, the update for the actor network parameters is
(7)$$\;\theta^{\mu }\leftarrow \theta^{\mu }+\alpha \delta_{t}\nabla_{\theta_{\mu }}\log\pi \left(a_{t}|s_{t},\;\theta^{\mu }\right)$$
where

$\pi \left(a_{t}|s_{t},\;\theta^{\mu }\right)\;$ is the policy, which represents the probability of taking action $a_{t}$ given state $a_{t}$.

To ensure stability during training, soft updates are performed on the target networks, gradually moving their parameters closer to those of the online networks:(8)$$\theta^{\mu^{\prime }}\leftarrow \tau \theta^{\mu }+\left(1-\tau \right)\theta^{\mu^{\prime }}$$
(9)$$\;\theta^{Q^{\prime }}\leftarrow \tau \theta^{Q}+\left(1-\tau \right)\theta^{Q^{\prime }}$$
where

τ is a small positive value representing the soft update rate.

The actor and critic networks work together, with the critic providing feedback via the TD error to guide the actor’s policy adjustments and to refine the value function. By leveraging both the local and shared experiences, the SEAC framework enables vehicles to improve their decision making for resource allocation in the 5G-NR-V2X networks. Soft updates maintain network stability by preventing drastic changes to the parameters of the target networks. This method allows the SEAC model to manage resource allocation effectively, minimizing collisions and congestion in high-traffic environments. By continuously sharing experiences and adjusting policies based on feedback, the framework adapts to varying traffic conditions and optimizes communication channels in dynamic V2X environments.

#### Multi-Agent MDP Model

The SEAC algorithm builds on the actor–critic framework in a multi-agent MDP model to address the limited information in decentralized systems. The MDP components include the state S representing the shared environment where all agents operate, actions $A_{i}$ taken by each agent that together form a joint action space A, and rewards $R_{i}$ given to each agent based on the state and actions. The system then transitions to the next state, which is influenced by the collective actions of all the agents. In SEAC, agents share their experiences—comprising state, action, and reward information—with other agents. This allows each agent to update its actor–critic model using both local and shared experiences, leading to improved decision making and more efficient resource allocation in future states [23].

The SEAC framework integrates actor and critic networks for effective learning. The actor network, parameterized by *ϕ*, selects actions based on the current state, while the critic network, parameterized by *θ*, evaluates these actions by estimating the value function.

Agents leverage both their own and others’ experiences through a shared experience pool. This off-policy learning approach adjusts the actor and critic loss functions to incorporate diverse interactions, enhancing the adaptability and resource allocation efficiency in dynamic environments.

In SEAC, agents share their experiences through a common experience pool, enhancing the learning efficiency. The goal is to determine policies $\pi =\left(\pi_{1},\pi_{2},\ldots ,\pi_{N}\right)$ that maximize the total reward over time T:(10)$$R^{i}=\sum_{t=0}^{T-1}\beta^{t}r_{t}^{i}$$
where β is the discount factor, and $r_{t}^{i}$ is the reward obtained by agent *i* at time *t*. The main objective for all agents is to maximize the expected reward:(11)Eπi′maxRi|π′,π−i ∀i∈πi
where $\pi_{-i}\;$ denotes the policies of all agents except *i*.

Each agent adjusts its strategy by minimizing losses in both the actor and critic networks, taking into account the partial observability of the environment. The actor network’s loss, which guides how it changes its policy, is defined as
(12)$$L\left(\varphi_{i}\right)=-\log\pi (a_{t}^{i}|o_{t}^{i},\varphi_{i})(r_{t}^{i}+\beta Q^{\prime }\left(s_{t+1},\;a^{\prime },\;\theta^{Q^{\prime }}\right)-Q\;\left(s,\;a,\;\theta^{Q}\right))$$
where

$\varphi_{i}\;$ are the actor’s network parameters;$Q\;\left(s,\;a,\;\theta^{Q}\right)$ is the value function computed by the critic network, $\mathrm{and}\;Q^{\prime }\left(s_{t+1},\;a^{\prime },\;\theta^{Q^{\prime }}\right)$ is the target value function.

The critic network’s loss function, which helps in evaluating actions, is
(13)$$L\left(\theta_{i}\right)={\left||Q\;(s,\;a,\;\theta^{Q}-y_{t}|\right|}^{2}$$

Here, $Q\;\left(s,\;a,\;\theta^{Q}\right)$ is the online value network, and $\;y_{t}$ is the target value network.

The target value $y_{t}$ is calculated using
(14)$$y_{t}=r_{t}^{i}+\beta Q^{\prime }\left(s_{t+1},\;a^{\prime },\;\theta^{Q^{\prime }}\right)$$
where

$r_{t}^{i}$ is the reward obtained by agent i at time t;β is the discount factor;$Q^{\prime }\left(s_{t+1},\;a^{\prime },\theta^{Q^{\prime }}\right)\;$ is the target value for the next state–action pair.

In SEAC, each agent not only learns from its own experiences but also incorporates experiences from other agents. This shared experience is considered “off-policy” data. The actor and critic losses are extended to include the off-policy experiences as follows:

The actor loss with experience sharing is
(15)$$al=log\pi (a_{t}^{i}|o_{t}^{i},\varphi_{i})(r_{t}^{i}+\beta Q^{\prime }(s_{t+1},\;a^{\prime },\;\theta^{Q^{\prime }})-Q\;(s,\;a,\;\theta^{Q}))$$
(16)$$L_{se}\left(\varphi_{i}\right)=L\left(\varphi_{i}\right)-\lambda \sum_{k\neq i}\frac{\pi \left(a_{t}^{k}|o_{t}^{k},\varphi_{k}\right)}{\pi \left(a_{t}^{k}|o_{t}^{k},\varphi_{i}\right)}\times al$$
where

$\pi (a_{t}^{i}|o_{t}^{i},\varphi_{i})$ represents the probability of taking action $a_{t}^{i}$ given the observation $o_{t}^{i}\;$ and the actor network parameters $\varphi_{i}$;$r_{t}^{i}$ is the immediate reward received by agent i at time t;β is the discount factor;$Q^{\prime }\left(s_{t+1},\;a^{\prime },\;\theta^{Q^{\prime }}\right)$ is the target Q-value at the next state $s_{t+1}$ using the next action $a^{\prime }$ and the target network parameters $\theta^{Q^{\prime }}$;$L\left(\varphi_{i}\right)$ is the standard actor loss for agent i without experience sharing;$\frac{\pi \left(a_{t}^{k}|o_{t}^{k},\varphi_{k}\right)}{\pi \left(a_{t}^{k}|o_{t}^{k},\varphi_{i}\right)}$ is the ratio of the policy probabilities of another agent k to that of agent i. This ratio adjusts how much influence the experiences of other agents should have on the current agent i’s policy.

The critic loss with experience sharing is
(17)$$L_{se}\left(\theta_{i}\right)=L\left(\theta_{i}\right)-\lambda \sum_{k\neq i}\frac{\pi \left(a_{t}^{k}|o_{t}^{k},\varphi_{k}\right)}{\pi \left(a_{t}^{k}|o_{t}^{k},\varphi_{i}\right)}{\left||Q\;\left(s,\;a,\;\theta^{Q}\right)-y_{t}|\right|}^{2}$$
where

$L\left(\theta_{i}\right)$ is the standard critic loss for agent i without experience sharing;$||Q\;\left(s,\;a,\;\theta^{Q}\right)-y_{t}||$ is the squared error between the predicted Q-value $Q\;\left(s,\;a,\;\theta^{Q}\right)$ and the target value $y_{t}$. This is a common loss function for training critic networks in reinforcement learning.

The hyperparameter λ adjusts how much influence the shared experiences have. The loss functions for the actor and critic are modified to include contributions from other agents, which are controlled by the hyperparameter λ. This approach ensures that the policy and value functions are updated not only based on the agent’s own experience but also by considering the experiences of other agents in the network [24,25].

By continuously updating policies and value functions based on real-time feedback, the SEAC ensures efficient and responsive resource allocation. The experience sharing mechanism allows agents to adjust their strategies, leading to better handling of traffic variability and ensuring that resources are allocated effectively without causing congestion or wastage.

## 4. Simulation Environments

The operation and performance of the SB-SPS, DS, and SEAC schemes were evaluated using the WiLabV2X simulator. This simulator provides an accurate implementation of 5G-NR-V2X Mode 2, adhering closely to the 3GPP specifications and evaluation guidelines.

For artificial intelligence (AI)-based allocation, we extended the existing simulation environment by a deep reinforcement learning (DRL) environment that executes the AI-based scheduler [26,27,28]. In this section, we discuss selected aspects of the evaluation methodology and the simulator implementation design [29,30,31,32].

### 4.1. AI-Driven Resource Management

In the simulation environment for evaluating AI-based radio resource allocation, the WiLabV2X simulator is employed, supporting the unmanaged mode for Cellular V2X communication. This simulator is primarily utilized to benchmark the performance of existing resource allocation techniques under unmanaged network conditions. For the AI-based resource allocation aspect, custom coding is developed independently, leveraging advanced deep reinforcement learning methodologies.

The dataset, encompassing parameters such as the distance to neighboring vehicles, forecast indicators, and radio resource occupancy, is meticulously prepared and imported. These parameters define the states crucial for training the reinforcement learning model. The state space incorporates factors like the number of radio resources occupied and the specific actions taken. The reward structure is designed to reflect the efficiency and effectiveness of resource usage, providing positive reinforcement for optimal allocations and penalties for suboptimal decisions.

### 4.2. Model Architecture

The core of the AI-based approach is a GRU model with two layers, which captures temporal dependencies and makes accurate future state predictions. The model architecture includes convolutional layers for feature extraction, GRU layers for sequential data processing, and fully connected layers for the final decision making.

The model trains and tests using a dataset split, with the training focused on optimizing parameters to reduce loss functions, such as the mean absolute percentage error (MAPE). The training process includes multiple epochs, with the model’s performance evaluated on both training and testing datasets to ensure generalization.

### 4.3. Simulation Parameters

The study is conducted on a simulated 2 km long highway segment with 3 lanes in each direction, totaling 6 lanes. Vehicles in the simulation maintain a mean speed of 140 km/h, with a standard deviation of 3 km/h. The system operates under the 5G-NR-V2X network with an OFDM numerology set to μ = 1, corresponding to a subcarrier spacing (SCS) of 15 kHz and a symbol time (ts) of 0.5 ms. The traffic type is aperiodic, with variable data size. The 5G-NR-V2X Mode 2 radios are configured to function within a 20 MHz channel in the 5.9 GHz ITS band, using QPSK modulation. Subchannels are organized into 12 resource blocks (RBs) each, resulting in 4 available subchannels per time slot. The transmission power is adjustable between 0 and 23 dBm, while the receiver sensitivity is set to −103.5 dBm. The simulation accounts for a proximity broadcast range of 500 m, with the resource reservation interval (RRI) set to 100 ms. The operational mode remains unmanaged, with an initial RSRP threshold of −128 dBm and a retention probability (P) of 0 for the SB-SPS scheme. These settings are detailed in Table 3.

The evaluation in this study utilizes a variety of V2X message types and simulation parameters to assess the performance of different scheduling mechanisms [28]. The key message types considered are Cooperative Awareness Messages (CAMs) and Collective Perception Messages (CPMs), as illustrated in Table 4. CAMs, defined in ETSI TS 103 900 V2.0.0 (July 2022), provide continuous updates on vehicle status and kinematics, typically at a frequency of 1–10 Hz with a payload size of around 400 bytes. CPMs, described in ETSI TS 103 324 V2.1.1 (Jun 2023), enable the sharing of sensor-based environmental perceptions, including detected and classified objects. These messages are also transmitted at a frequency of 1–10 Hz, with payloads of up to 1000 bytes [29].

### 4.4. Performance Metrics

In this study, the performance of the SEAC reinforcement learning technique is compared with traditional SB-SPS and DS schemes using the following metrics:f
Packet reception ratio (PRR): The PRR measures the percentage of correctly received and decoded transmission bursts (TBs) compared to the total transmitted TBs. It is evaluated as a function of the distance (D) between transmitting and receiving vehicles, providing insight into the reliability of communications over varying distances.
(18)$$PRR\;\left(D\right)=\frac{N_{rec\;}\left(D\right)}{N_{trans}\left(D\right)}\times 100$$
where$N_{rec\;}\left(D\right)$ is the number of TBs correctly received and decoded at distance D;$N_{trans}\left(D\right)$ is the total number of TBs transmitted at distance D.
Radio resource utilization (RRU): This metric evaluates the efficiency with which the communication system uses the available radio spectrum. It reflects the effectiveness of resource management in maintaining high network performance and minimizing wasted resources.(19)$$RRU=\frac{N_{used\;}}{N_{available}}\times 100$$
where$N_{used\;}$ is the number of radio resources that were successfully used for communicating;$N_{available}$ is the total number of radio resources available for use.
Probability of resource collision (PRC): This metric assesses the likelihood of multiple vehicles selecting the same radio resources simultaneously, causing communication collisions. A high PRC indicates frequent resource contention, leading to degraded network performance. Minimizing the PRC is essential for efficient spectrum utilization and reliable communication, especially in dense traffic scenarios.(20)$$RRU=\frac{N_{collisions\;}}{N_{total\_attempts}}\times 100$$
where$N_{collisions\;}$ is the number of instances where multiple vehicles selected the same resource, causing a collision;$N_{total\_attempts}$ is the total number of attempts made by vehicles to access the resources.


In addition to these metrics, the effectiveness of the SEAC technique is further demonstrated through training graphs, illustrating the model’s learning progress. These graphs provide a visual representation of how well the SEAC model improves its resource allocation decisions over time, showcasing its potential advantages over traditional methods.

## 5. Performance Evaluation and Analysis

The results discussed in this section are derived from the deployment of both rule-based and AI-based algorithms in the simulator, as outlined in Section 3. The relevant simulation parameters are detailed in Table 3. To demonstrate the effectiveness of the AI-based approach over the rule-based radio resource scheduling in Cellular V2X environments, three categories of results are presented: (i) measurable parameters at the access layer of the Cellular V2X system; (ii) performance of the GRU layers; and (iii) learning curves of the AI training model.

### 5.1. Key Performance Indicators (KPIs)

The optimization goal of this study is clearly defined in terms of key performance indicators (KPIs), which include the probability of resource collision (PRC), the packet reception ratio (PRR), and the radio resource utilization (RRU). These KPIs directly measure the performance improvements brought about by the proposed SEAC technique.

All the results in this study demonstrate that the Shared Experience Actor–Critic (SEAC) method with *N* steps (*N* = 10) consistently outperforms traditional methods like DS and SB-SPS. These trends are consistent with findings from other studies [2], where SEAC has shown improvements in reducing the PRC and enhancing the PRR. However, unlike the method in [2], which focuses solely on immediate rewards, our approach uses *N* steps to optimize both short-term and long-term rewards. This allows for more informed decision making and better resource management, leading to superior performance, particularly in high-density traffic scenarios.

#### 5.1.1. Probability of Resource Collision (PRC)

It is to be noted that all vehicles in the simulation operate in the unmanaged Mode of 5G-NR-V2X Mode 2. Figure 5 shows the PRC under an aperiodic traffic environment. The graph compares the SEAC reinforcement learning technique with traditional methods like SB-SPS and DS.

At the beginning, all the techniques show a similar PRC at low channel busy ratios (CBRs), indicating comparable effectiveness in low-traffic scenarios. However, as the CBR increases, traditional methods like SB-SPS and DS quickly experience a sharp rise in the PRC, leading to higher collision rates. In contrast, the SEAC, particularly in the 10-step simulation, maintains a lower PRC even at higher CBR levels. The SEAC model with 10 steps refers to the reinforcement learning model using a 10-step lookahead strategy. This approach allows the model to make more informed decisions by considering the potential outcomes of actions over the next 10 time steps, optimizing the resource allocation and collision avoidance.

The model considers the potential outcomes and rewards over the next 10 steps before updating its policy. This allows the SEAC model to make more informed decisions by anticipating the future impact of its actions. The cumulative reward is calculated over N steps, which helps in reducing variance and provides a smoother learning process. This approach balances the immediate rewards and the long-term benefits, making the learning process more stable and robust. This demonstrates SEAC’s superior ability to manage resources and prevent collisions under increasing channel congestion, offering a more resilient solution in high-density traffic scenarios.

#### 5.1.2. Packet Reception Ratio (PRR)

The reliability of the communication system used in Cellular Vehicle-to-Everything (V2X) networks, specifically in sidelink broadcast communication, is assessed by looking at how often messages are successfully received. This is evaluated in terms of the PRR. Figure 6 illustrates the PRR as a function of the number of vehicles per mile per lane, comparing the SEAC approach, including its 10-step variant, with traditional methods such as DS and SB-SPS. The key difference between SEAC and SEAC with 10 steps lies in how rewards are accumulated: SEAC generally optimizes actions based on immediate feedback, while SEAC with 10 steps accumulates rewards over a fixed 10-step window, allowing for more informed long-term decision making.

As the vehicular density increases, all methods show a decline in the PRR. However, SEAC, particularly with the 10-step configuration, consistently achieves a higher PRR compared with DS and SB-SPS. SEAC’s superior performance is evident from its ability to maintain near-perfect reception at low densities and a slower decline in the PRR as the density increases. In contrast, DS and SB-SPS experience more significant reductions in the PRR, indicating less effective resource management.

#### 5.1.3. Radio Resource Utilization (RRU)

Figure 7 compares the RRU across different vehicular densities—low, medium, and high—using various scheduling methods in a 5G-NR-V2X Mode 2 environment with aperiodic traffic and variable data sizes. It evaluates the performance of DS, SB-SPS, and the proposed SEAC with an N-step approach, where the value of N is set to 10.

The SEAC method shows consistently higher RRU, with utilization rates of 75.2%, 70.21%, and 64.36% for low, medium, and high vehicular densities, respectively. This indicates superior efficiency in resource management compared with DS and SB-SPS, which exhibit a noticeable decline in RRU as the vehicular density increases. The SEAC’s use of GRUs for predictive modeling allows for real-time adjustments in resource allocation, effectively optimizing bandwidth usage and managing the variability in data traffic demands. This ensures reliable and efficient communication in V2X networks, particularly in dynamic and complex scenarios.

### 5.2. GRU Performance

The GRU model facilitates efficient resource reservation in the V2X network by accurately predicting future vehicle positions. This process begins with the GRU layers analyzing historical position data, identifying patterns and sequences in the movements of vehicles over time. By capturing these temporal dependencies, the model can generate forecasts regarding the future positions of both the ego vehicle and its neighboring vehicles. These predictions are crucial for anticipating changes in the network, such as whether vehicles will move closer to or further from the ego vehicle, or if they will enter or exit the ego-vehicle sensor’s field of view (FOV). Such foresight is vital for planning the allocation of communication resources.

The model dynamically adjusts the resource reselection counter ($C_{resel}$), determining the duration for which channels are reserved. Accurate predictions help avoid both the over-allocation and under-allocation of resources, optimizing bandwidth use. In essence, the GRU model ensures efficient and reliable data transmission by enabling precise and adaptive resource management.

#### 5.2.1. GRU Model Architecture

The architecture details shown in Table 5 indicate a two-layer GRU network followed by a series of dense layers. The GRU layers contain 150 and 300 units, respectively, enabling the model to capture complex temporal patterns in resource demand and availability.

The model also includes a sigmoid activation function at the output layer, which is suitable for determining the optimal allocation or reallocation of resources in the V2X communication scenario. This setup allows the system to efficiently predict and manage resource allocation decisions.

#### 5.2.2. A2C CNN-GRU Model

The Actor–Critic Central Neural Network (A2C CNN-GRU) model, as detailed in Table 6, integrates convolutional, GRU, and dense layers to optimize radio resource management in V2X networks. The model’s critic component evaluates the value of state–action pairs, which helps to allocate resources like frequency and bandwidth efficiently. The consistent decrease in critic loss values reflects the improving accuracy in these evaluations.

The actor component maintains stable policy decisions, as indicated by the low actor loss values in Figure 9, ensuring effective decisions on channel allocation and data prioritization. The GRU layers capture temporal dependencies, allowing the model to predict future network states and to adjust resource allocations accordingly. This capability helps manage dynamic changes in traffic and demand, ensuring optimal use of the radio spectrum and enhancing communication reliability and efficiency.

The two provided graphs illustrate the model’s predictions for the distance to neighboring vehicles, which are normalized and compared across actual, training, and testing datasets. The *x*-axis represents time steps, while the *y*-axis denotes the distance in meters. In Figure 8a,b, both graphs present the normalized distance to neighboring vehicles in meters over 500 time steps for three datasets: actual, training, and testing. The blue line represents the actual dataset, which shows a rapid increase in distance that stabilizes at around 60 m after approximately 100 time steps. The red line, indicating the training dataset, closely follows the actual dataset, signifying that the model has been trained effectively and captures the underlying trend accurately. Similarly, the green line for the testing dataset closely follows the actual dataset, demonstrating that the model generalizes well to unseen data, maintaining a consistent prediction trend.

These results suggest that the model is robust, effectively trained, and capable of generalizing well to new data, making it reliable for predicting the normalized distance to neighboring vehicles over time. The alignment of the training and testing datasets with the actual measurements indicates that the model accurately captures the movement patterns of vehicles. This accuracy is crucial for anticipating changes in traffic flow and managing resource allocation.

### 5.3. Learning Curves

The learning performance is illustrated through two key graphs. The first graph (Figure 9a) shows the loss curves for both the critic and actor components. The critic loss consistently decreases over 100 epochs, indicating enhanced accuracy in evaluating state-action pairs, while the actor loss remains low and stable, reflecting effective policy learning. The second graph (Figure 9b) presents the reward progression, which indicates the model’s success in maximizing network efficiency. The rewards steadily increase, demonstrating the model’s improving performance in optimizing resource allocation. This upward trend signifies the model’s growing ability to enhance V2X communication reliability and minimize resource wastage and collision.

The convergence of the proposed DRL method is demonstrated in Figure 9, where we observe a significant decrease in critic loss over the course of 100 epochs, stabilizing near zero. The actor loss remains consistently low, further supporting the convergence of the model. Additionally, the rewards graph shows a steady increase, indicating that the model is learning effectively and approaching optimal performance. The reduction in losses and the increase in rewards over the epochs confirm that the model has converged successfully based on the number of trials conducted.

## 6. Conclusions

This study explores the challenges of resource allocation in 5G-NR-V2X Mode 2 networks, particularly under aperiodic traffic conditions, and proposes the SEAC technique as a solution. Our findings indicate that traditional approaches, such as SB-SPS and DS, exhibit notable limitations, especially in high-density scenarios, where resource collisions and inefficiencies are pronounced. Under high-traffic conditions, a SEAC model with N steps achieves an RRU of approximately 70.36%, representing a 60.01% improvement over SB-SPS and a 32.12% improvement over DS. SEAC also maintains a POR of around 0.85 as the vehicular density increases, which is 10–15% higher than DS and 20–25% higher than SB-SPS. Additionally, SEAC demonstrates a PRC that is 15–20% lower than DS and 20–30% lower than SB-SPS in high-density environments. The SEAC method, leveraging reinforcement learning in a multi-agent environment, shows a clear advantage in optimizing resource allocation. These results underscore SEAC’s potential to enhance the reliability and efficiency of V2X communications, which are critical for the advancement of autonomous driving technologies. Future research can explore the combination of SEAC with other AI-based techniques, as well as its application in diverse vehicular scenarios, including urban environments with complex traffic dynamics. By addressing these challenges, the SEAC technique has the potential to play a pivotal role in shaping the future of ITSs.

## Figures and Tables

**Figure 1 sensors-24-06769-f001:**
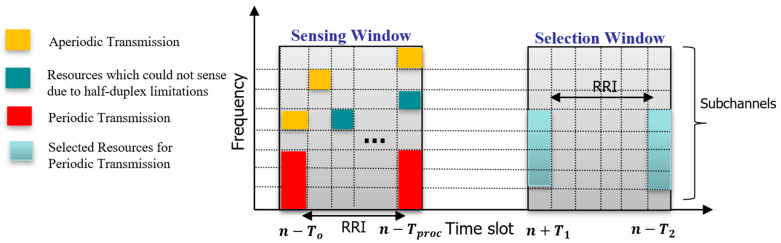
Resource selection procedure in 5G-NR-V2X Mode 2.

**Figure 2 sensors-24-06769-f002:**
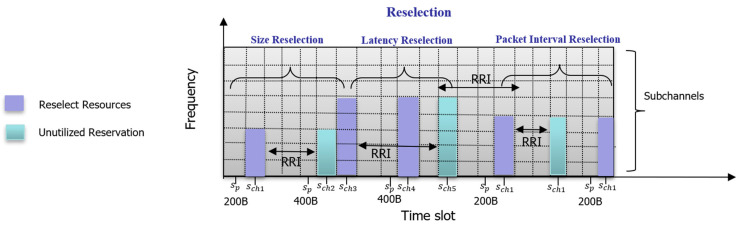
Impact of varying traffic types in 5G-NR-V2X Mode 2.

**Figure 3 sensors-24-06769-f003:**
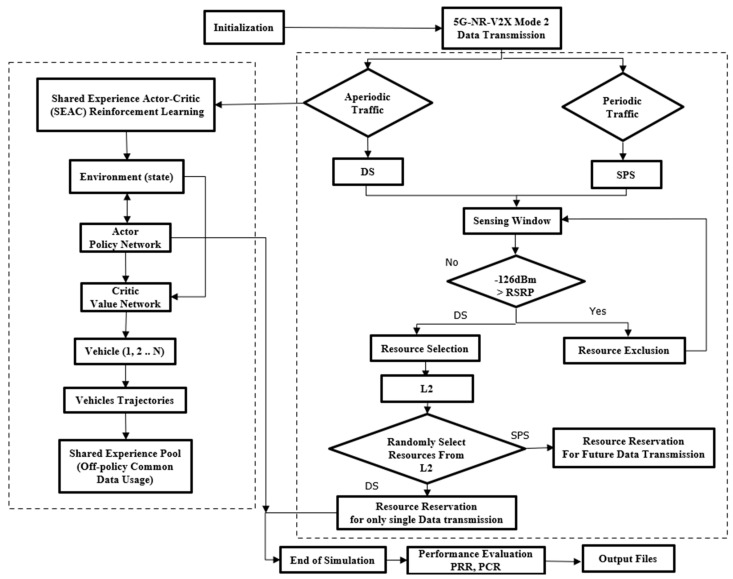
Framework for resource allocation strategies in 5G-NR-V2X Mode 2.

**Figure 4 sensors-24-06769-f004:**
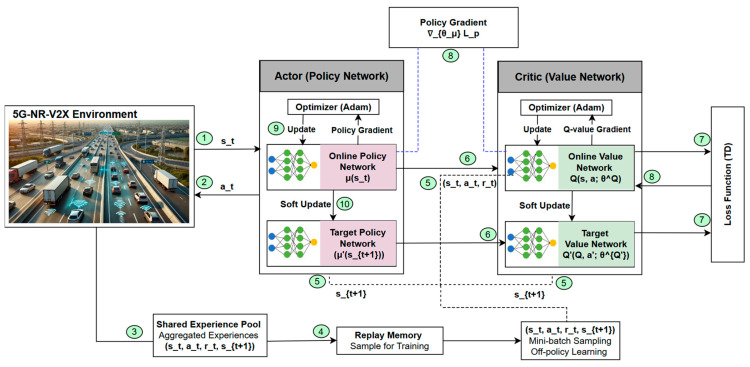
Network structure and flow of the SEAC algorithm.

**Figure 5 sensors-24-06769-f005:**
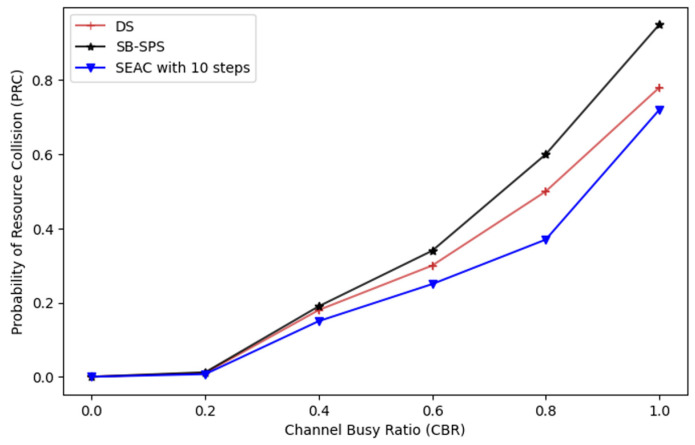
Probability of resource collision (PRC) relative to the channel busy ratio (CBR).

**Figure 6 sensors-24-06769-f006:**
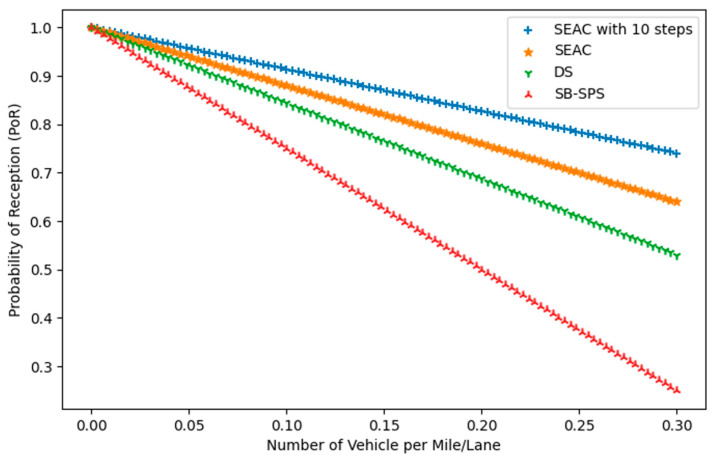
Probability of reception (PoR) vs. the number of vehicles per mile/lane.

**Figure 7 sensors-24-06769-f007:**
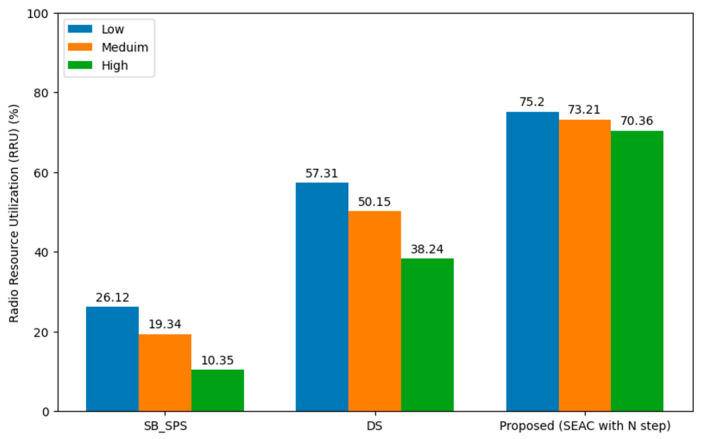
Radio resource utilization (RRU) across varying vehicular densities.

**Figure 8 sensors-24-06769-f008:**
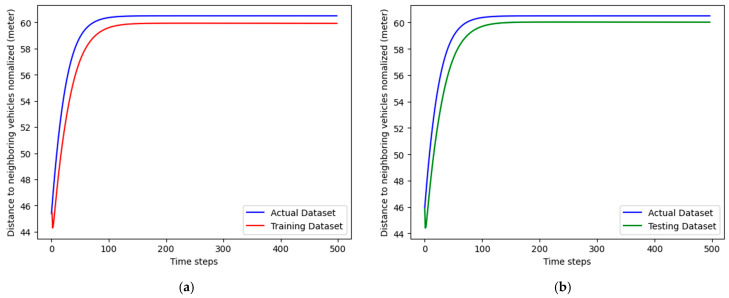
A2C model training and testing datasets. (**a**) A2C Model Performance on Training Dataset. (**b**) A2C Model Performance on Testing Dataset.

**Figure 9 sensors-24-06769-f009:**
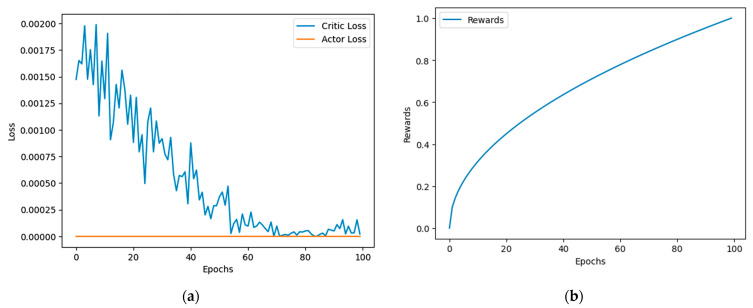
A2C model loss analysis. (**a**) Actor vs. critic loss. (**b**) Reward performance.

**Table 1 sensors-24-06769-t001:** Reselection types in 5G-NR-V2X Mode 2.

	Counter Reselection	Size Reselection	Latency Reselection
SB-SPS	$C_{resel}$ = 0, depends on *P*	TB size larger than reservation size	RRI > PDB
DS	Every TB	Never	Never

**Table 2 sensors-24-06769-t002:** Comparison of related schemes.

Work	AI-Based Resource Allocation	SB-SPS	DS	Periodic Traffic	Aperiodic Traffic	Highway Scenario	Urban Scenario
[1]	✗	✓	✓	✓	✓	✓	✗
[2]	✓	✓	✗	✓	✓	✗	✗
[5]	✗	✓	✗	✓	✗	✗	✓
[9]	✓	✓	✗	✓	✓	✓	✗
[12]	✗	✓	✗	✓	✓	✗	✗
[13]	✗	✓	✗	✓	✓	✗	✓
[24]	✗	✓	✗	✓	✓	✓	✗
[30]	✗	✓	✓	✓	✓	✓	✗

**Table 3 sensors-24-06769-t003:** Simulation parameters.

Parameter	Values
Network	5G-NR-V2X Mode 2
Highway Segment Length	2 km
Lanes Per Direction	3
Initial RSRP Threshold	−128 dBm
Mean Vehicle Speed	140 km/h
Standard Deviation of Speed	3 km/h
Reselection Counter	[5, 15]
Data Traffic Type	CAM and CPM broadcast
Subcarrier Spacing (SCS)	15 kHz
Operational Mode	Unmanaged
Channel Bandwidth	20 MHz
Operating Band	5.9 GHz ITS band
Modulation Scheme	QPSK
Transmission Power	0–23 dBm
Receiver Sensitivity	−103.5 dBm
Resource Reservation Interval	100 ms
Proximity Broadcast Range	500 m
Traffic Type	Aperiodic

**Table 4 sensors-24-06769-t004:** CAMs and CPMs.

Message Type	ETSI Standard	Scope	Traffic Characteristic
CAM—Cooperative Awareness Message	TS 103900 V2.0.0 (July 2022)	CAMs periodically broadcast vehicle status, like position and speed, to improve situational awareness.	1–10 Hz, ~400 bytes
CPM—Collective Perception Message	TS 103 324 V2.1.1 (Jun 2023)	CPMs share sensor data about detected objects, aiding in collective traffic perception and management.	1–10 Hz, up to 1000 bytes

**Table 5 sensors-24-06769-t005:** GRU model architecture.

Layer (Type)	Output Shape	Parameters
GRU-1	[−1, 1, 150]	0
GRU-2	[−1, 2, 300]	0
Flatten-3	[−1, 300]	0
Linear-4	[−1, 150]	45,150
Linear-5	[−1, 75]	11,325
Linear-6	[−1, 1]	76
Sigmoid-7	[−1, 1]	1

**Table 6 sensors-24-06769-t006:** A2C model architecture.

Layer (Type)	Output Shape	Parameters
Conv1d	[−1, 32, 6]	128
SiLU	[−1, 32, 6]	0
MaxPool1d	[−1, 32, 3]	0
Conv1d	[−1, 64, 5]	6208
SiLU	[−1, 64, 5]	0
MaxPool1d	[−1, 64, 2]	0
GRU	[[−1, 64, 8], [−1, 64, 8]]	0
Flatten	[−1, 512]	0
Dropout	[−1, 512]	0
Linear	[−1, 64]	32,832
ReLU	[−1, 64]	0
Dropout	[−1, 64]	0
Linear	[−1, 32]	2080
ReLU	[−1, 32]	0
Linear	[−1, 2]	66
Linear	[−1, 32]	1056
ReLU	[−1, 32]	0
Linear	[−1, 1]	33

## Data Availability

Data are contained within the article.

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
