# Peer review of "The Shared Experience Actor–Critic (SEAC) Approach for Allocating Radio Resources and Mitigating Resource Collisions in 5G-NR-V2X Mode 2 Under Aperiodic Traffic Conditions"

_sensors, 2024, doi:10.3390/s24206769_

Round 1

Reviewer 1 Report

Comments and Suggestions for Authors

The paper is very well written in all aspects including mathematical details. My only concern is the validity of the results presented in Figures 5 and 6. How can we trust these results?

Author Response

I attached the file below for your reference. Please consider it.

Reviewer 2 Report

Comments and Suggestions for Authors

This paper investigates radio resource allocation and resource collision mitigation in 5G-NR-V2X mode 2 by shared experience actor critic approach. My comments are provided as follows:

1. The abstract should be improved. The introduction of the background takes up too much space, and the main work of this paper should be emphasized.

2. Page 2, “… mode 1 (centralized) and Mode 2 (decentralized)”. Why do you use lowercase letter for model 1, but uppercase letter for Model 2?

3. The manuscript has three authors. However, in line 182, the authors stated that “In my work, I address …”. This seems contradictory.

4. Line 230, “s_t” should be changed to “st”.

5. For eq. (4), please explain the meaning of “N-(1)1”.

6. In Fig. 6, what is the difference between SEAC and SEAC with 10 steps?

7. In Fig. 7, for the proposed SEAC with an N-step approach, what is the value of N?

8. For Fig. 9, the titles of subfigures (a) and (b) should be added.

Author Response

I have attached the file below for your reference. Please consider it.

Reviewer 3 Report

Comments and Suggestions for Authors

1. The system model is not clear. What are the design constraints?

2. The optimization goal should be specified in terms of KPI.

3. The convergence of the proposed DRL method should be analyzed together with fig.9.

4. There are some related works can be included for comparison such as follow:

[1] "Joint Interference Alignment and Power Control for Dense Networks via Deep Reinforcement Learning," in IEEE Wireless Communications Letters

[2] "Reputation Management for Consensus Mechanism in Vehicular Edge Metaverse," in IEEE Journal on Selected Areas in Communications

Author Response

(The authors gave the same response as above.)

Round 2

Reviewer 2 Report

Comments and Suggestions for Authors

I have no more comments.

Author Response

I have attached below the response letter for your reference. Please consider it

Reviewer 3 Report

Comments and Suggestions for Authors

The reviewer appreciate the effort on revising the manuscript.

The author is encouraged to replace the figures of simulation results with vector graph.

Author Response

(The authors gave the same response as above.)
